# Tocilizumab in Combination with Corticosteroids in COVID-19 Pneumonia: A Single-Centre Retrospective Controlled Study

**DOI:** 10.3390/biomedicines11020349

**Published:** 2023-01-26

**Authors:** Zsófia Kardos, Miklós Szabó, Zsuzsanna Baráth, Ágnes Miksi, Csaba Oláh, Ádám Kozma, József A. Gergely, Eszter Csánky, Zoltán Szekanecz

**Affiliations:** 1Departments of Rheumatology, Borsod Academic County Hospital, H-3529 Miskolc, Hungary; 2Faculty of Health Sciences, University of Miskolc, H-3515 Miskolc, Hungary; 3Departments of Pulmonology, Borsod Academic County Hospital, H-3529 Miskolc, Hungary; 4Departments of Neurosurgery, Borsod Academic County Hospital, H-3526 Miksolc, Hungary; 5Departments of Institutional Pharmacy, Borsod Academic County Hospital, H-3526 Miskolc, Hungary; 6Department of Rheumatology, Faculty of Medicine, University of Debrecen, H-4032 Debrecen, Hungary

**Keywords:** COVID-19, IL-6 receptor, tocilizumab, pneumonia, efficacy

## Abstract

**Introduction:** Interleukin 6 receptor inhibition by tocilizumab (TCZ) has been effectively used worldwide for the treatment of multisystem inflammatory syndrome (MIS) associated with COVID-19. In this single centre study, we compared the outcome of COVID-19 pneumonia in TCZ-treated vs. untreated (control) patients. We wished to compare TCZ administration in the general ward vs. in the intensive care unit (ICU). We also studied the role of a consulting rheumatologist in the management of severe COVID-19 pneumonia. **Patients and methods:**
In our patients, COVID-19 pneumonia was confirmed by SARS-CoV-2 PCR, chest X-ray, and CT. We compared patients selected for TCZ treatment with TCZ-untreated age- and sex-matched controls. All patients received corticosteroids. In the TCZ-treated group, patients received one or two doses of TCZ 8 mg/kg IV in combination with corticosteroids. We recorded age, sex, symptom duration, oxygen saturation (SaO_2_), partial arterial oxygen pressure (PaO_2_), total white blood cell (WBC), absolute neutrophil, absolute lymphocyte and platelet counts, CRP, ferritin, IL-6, LDH, procalcitonin (PCT), and D-dimer. The primary outcome parameters were the need for ICU, ventilation, death, and time of hospitalisation. **Results:** Altogether, 104 patients, 52 TCZ-treated and 52 TCZ-untreated, were included in this study. At baseline, the TCZ-treated patient group indeed had more pronounced COVID-19-related MIS compared to controls. Consultation with a rheumatologist was performed in 60% vs. 40% of cases. Nineteen patients (37%) received one, while 33 (63%) received two TCZ doses. TCZ was administered to 28 patients (54%) in the general ward and to 24 (46%) in the ICU. TCZ treatment was found to be safe in our COVID-19 pneumonia patients. TCZ treatment favourably influenced MIS biomarkers, and was associated with better clinical outcomes compared to controls. Patients receiving TCZ treatment in combination with corticosteroids already in the general ward exerted much better outcomes than those treated in the ICU. Consultation with a rheumatologist also improved outcome. **Conclusions:** We successfully used TCZ in combination with corticosteroids in Hungarian COVID-19 pneumonia patients. We pointed out the importance of early treatment already in the general ward, and the involvement of a rheumatologist in making treatment decisions.

## 1. Introduction

Since the onset of the Coronavirus Disease 19 (COVID-19) pandemic, we have learned a lot about the development, clinical signs and symptoms, diagnosis, treatment, course, and outcome of the disease [1,2]. The course of COVID-19 includes multiple stages, which also determine the indicated treatment strategy [1,2,3]. Stage 1 is the period of early viral infection with fever, respiratory or gastrointestinal symptoms, and lymphopenia. Stage 2 is the pulmonary phase, which includes the non-hypoxemic Stage 2a and the hypoxemic Stage 2b. Finally, Stage 3 is the phase of multisystemic inflammatory syndrome (MIS), occasionally accompanied by the cytokine storm as a pathogenetic feature [1,2,3]. As anti-inflammatory agents are most effective during MIS, this should be confirmed by clinical, imaging, and laboratory markers [1,4,5,6,7]. Among laboratory biomarkers, increased serum C-reactive protein (CRP), ferritin, D-dimer, and interleukin 6 (IL-6) levels, as well as leucocytosis, neutrophilia, lymphocytopenia, elevated neutrophil-to-lymphocyte ratio, and thrombocytopenia have been associated with MIS in Stages 2b-3, and also with the outcome of COVID-19 [5,6,8]. 

As COVID-19-associated MIS is in many ways similar to systemic inflammatory rheumatic diseases, numerous drugs used for the treatment of rheumatic conditions have been repurposed in COVID-19 [7,9,10]. These agents include corticosteroids, anti-cytokine biologics, as well as JAK inhibitors [7,9,10]. Among these compounds, the IL-6 receptor (IL-6R) inhibitor tocilizumab (TCZ) has been effective in several randomized clinical trials (RCT) in Stage 2b-3 COVID-19 [11,12,13,14]. In these studies, TCZ was effective in severe cases requiring intensive care unit (ICU) admission and ventilation. In these patients, TCZ improved survival and the chance of hospital discharge [11,12,13]. There have been numerous studies; therefore, we just mention a few key investigations. In the REMAP-CAP study, TCZ was so effective within two days of ICU referral that the study was prematurely terminated [12]. In the CHIC study conducted by EULAR, TCZ in combination with a corticosteroid improved the clinical picture and survival in patients initially treated with corticosteroids only, but who did not show adequate response [15]. The largest COVID-19 therapeutic study to date (RECOVERY; 4116 patients) included patients requiring invasive ventilation, non-invasive oxygen therapy, or none of these (control group). At baseline, 82% of the patients received corticosteroids. In hypoxic Stage 2b-3 patients requiring hospitalisation, TCZ in comparison to standard care significantly reduced invasive mechanical ventilation or mortality, and improved the chance of hospital discharge within 28 days. TCZ also decreased the need for invasive ventilation [13]. Finally, the WHO REACT consortium performed a meta-analysis of studies on anti-IL-6 receptor blocking agents, TCZ and sarilumab [16]. Data coming from 10,930 patients were evaluated. In this prospective meta-analysis of clinical trials, the administration of IL-6 receptor antagonists, compared with usual care or placebo, was associated with lower 28-day all-cause mortality [16]. 

On the other hand, TCZ was not effective in patients with mild-to-moderate COVID-19 not requiring ICU admission [14,17,18,19]. In the first large phase 3 COVACTA trial conducted by the sponsor, neither the primary (clinical improvement) nor the secondary endpoints (death) were met. Only a moderate improvement in terms of hospital discharge was observed [20,21]. Based on the subsequent evaluation, it was found that patient enrolment was only based on the fact of having COVID-19 and the need for oxygen; biomarkers suitable for confirming MIS were not applied [21]. 

In most successful studies, TCZ was combined with corticosteroids. In these studies, TCZ in combination with corticosteroids was more effective than TCZ monotherapy in terms of both survival and the need for invasive ventilation [15,16,22]. 

With respect to biomarkers suitable for monitoring TCZ therapy, in one study, the cut-off value of CRP for the effectiveness of TCZ was found to be 3.5 mg/dL. If CRP was >3.5 mg/dl, TCZ reduced mortality by 35%, while it had no pronounced effect in the case of CRP ≤ 3.5 mg/dl. [23]. It should also be noted that TCZ transiently increases circulating IL-6 levels (due to the competitive binding to the IL-6 receptor); therefore, the determination of serum IL-6 concentration is only recommended at baseline, before the initiation of treatment. For this reason, the regular assessment of serum IL-6 levels is not suitable for monitoring therapy [24]. For follow-up, in addition to clinical symptoms and radiological findings, changes in CRP, ferritin, D-dimer, cTn, and LDH can be used [5]. 

Thus, TCZ is recommended in patients with confirmed MIS who do not respond to corticosteroids [12,13,15], but not in the early stages of COVID-19, nor in the absence of significant inflammation [17,18].

The management of COVID-19-associated MIS, in addition to intensive care specialists, requires professionals experienced in the use of anti-cytokine biologics, such as rheumatologists. Our hospital has been serving as a regional COVID-19 centre. During the second and third waves of the pandemic, we introduced the use of TCZ to patients treated at our COVID departments. In this study, we compared patients with COVID-19 pneumonia treated with TCZ with those not receiving TCZ. We assessed outcomes (need for ICU admission, need for ventilation, and death), as well as biomarkers of MIS. We compared patients receiving TCZ already in the general ward with those admitted to ICU. We also wished to determine the effect of a consultation with a rheumatologist on clinical outcomes. Our aim was to, after numerous international studies, perform the first Hungarian comparative study on the use of TCZ in addition to corticosteroids in severe COVID-19 pneumonia.

## 2. Patients and Methods

### 2.1. Patients

This single-centre, retrospective cohort study was conducted at the dedicated COVID-19 department of the Borsod Academic County Hospital, Miskolc, Hungary. Data from patients hospitalised for COVID-19 pneumonia between 16 October 2020, and 1 April 2021, were retrospectively analysed. Confirmation of SARS-CoV-2 infection was performed by the RT-PCR method from throat-swab specimens. Pneumonia was confirmed by radiological imaging performing chest CT or plain X-ray in all cases. We compared patients selected for TCZ treatment with TCZ-untreated age- and sex-matched controls (Table 1). All patients received antiviral treatment, either favipiravir or remdesivir or both. All patients also received corticosteroids intravenously (methylprednisolone—Solu-Medrol Pfizer^®^—40–80 mg twice daily, or dexamethasone—Dexa-ratiopharm^®^—6 mg once a day), a prophylactic dose of enoxaparine, low-dose aspirin, and mucolytics, as well as oxygen supplementation if needed, according to the national recommendations [3]. 

The clinical criteria for hospital discharge included absence of fever for at least 3 days, cessation or significant improvement of respiratory symptoms, as well as clear improvement of the radiological picture.

The Ethics Committee of the Borsod Academic County Hospital approved this study (BORS 04/2021). We conducted this study according to the Declaration of Helsinki.

### 2.2. Tocilizumab Dosing Regimen

In the TCZ-treated group, according to the recommended protocols, all patients received at least one dose of TCZ (8 mg/kg, max. 800 mg IV), which could be repeated once more within 24 h if required [3,17,19]. TCZ was administered in one 60 min infusion to patients receiving systemic corticosteroid treatment, and those in need of oxygen therapy or ventilation. If, after the first dose, the clinical symptoms do not improve or worsen, a second dose of 8 mg/kg may be administered after a minimum of 8 h [3,17,19].

### 2.3. Clinical, Laboratory, and Imaging Data Collection

We reviewed all clinical electronic medical records, and laboratory reports, as well as chest CT and X-ray images. We recorded age, sex, the duration of symptoms, oxygen saturation (SaO_2_) determined by pulse oximetry, and partial arterial oxygen pressure (PaO_2_) assessed by blood gas analysis. Laboratory investigations included total white blood cell (WBC), absolute neutrophil, absolute lymphocyte and platelet counts, as well as serum CRP, ferritin, IL-6, LDH, D-dimer, procalcitonin (PCT), and D-dimer. We calculated neutrophil-to-lymphocyte ratio in each patient. The use of corticosteroids and antiviral agents was also recorded. As rheumatologists are experts in using TCZ in rheumatoid arthritis, we also assessed how a consultation between the managing physician and a rheumatologist would influence clinical outcomes. 

Chest CT scans were performed using a single inspiratory phase in a 128 multi-detector CT scanner (SOMATOM Go Top, Siemens Healthineers, Germany). In order to minimize motion artifacts, patients were instructed on breath-holding; CT images were then acquired during a single breath-hold. For CT acquisition, the tube voltage was 90 kVp with automatic tube current modulation. From the raw data, 1 mm slices were reconstructed with a pulmonary Br 64 kernel and a mediastinal Br40 kernel (Siemens Healthineers, Germany) in all three planes. All thin-section CT images were reviewed at a window width and level of 400 and 40 HU and 1200 and −600 HU, respectively, for the mediastinum and lung parenchyma. We determined CT severity scores (CTSS) according to the protocol by Pan et al. [25], as described previously [26]. In brief, the extent of anatomic involvement was calculated in each of the 5 lobes. In each lobe, the absence of lobar involvement (0%) yielded to a score of 0, while <5%, 5–25%, 26–50%, 51–75%, and >75% involvement was scored as 1, 2, 3, 4, and 5, respectively. Thus, the individual scores of the five lobes resulted in a global score of 0 to 25.

All data were evaluated by two physicians (ZK, MS), and a third researcher (ZS) adjudicated any difference in interpretation between the two primary reviewers. 

### 2.4. Outcome Parameters

The primary outcome parameters were the need for intensive care, need for ventilation (invasive or non-invasive), and death. The time of hospitalisation was also recorded. We also compared when TCZ was administered in a general ward vs. in the ICU and correlated this with all other parameters.

### 2.5. Statistical Analysis

Statistical analysis was performed using the SPSS software v.29.0 (IBM, Armonk, NY, USA). Data are expressed as mean ± SD for continuous and case number plus percentages (*n*, %) for categorical variables. The distribution of continuous variables was determined by Kolmogorov–Smirnov test. Continuous variables were assessed by Mann–Whitney U tests. Nominal variables were compared by χ^2^ or Fisher’s exact test. Spearman’s analysis was used to test for correlations. *p* values <0.05 were considered significant in all tests mentioned above. 

## 3. Results

### 3.1. Characterization of Patients

Altogether, 104 patients, 52 TCZ-treated and 52 TCZ-untreated (group), were included in this study. Their main characteristics are included in Table 1. The full patient cohort included 35 men and 17 women in both groups. In the TCZ-treated and control groups, the mean age was 60.2 ± 9.6 years [range: 37–78 years] and 60.1 ± 9.8 years [range: 40–76 years], respectively. Disease duration (time from the first symptom to hospital admission) was 9.75 ± 3.66 days [range: 1–17 days] and 9.65 ± 3.46 days [range: 1–15 days], respectively. The TCZ-treated and control groups were matched for patient numbers, age, sex, symptom duration, radiological extent of lung involvement (CTSS), antiviral therapy, and corticosteroid use, as there were no significant differences in these parameters between the two groups (Table 1). 

With respect to laboratory biomarkers, at baseline, the TCZ-treated patient group indeed had more pronounced COVID-19-related MIS compared to controls. In the TCZ-treated group, baseline total WBC counts (*p* < 0.001), absolute neutrophil counts (*p* < 0.001), neutrophil-to-lymphocyte ratio (*p* = 0.005), serum IL-6 (*p* = 0.003), ferritin (*p* = 0.026), and LDH levels (*p* = 0.020) were higher compared to controls (Table 1). 

Before determining the indication for TCZ therapy, consultation with a rheumatologist (ZK) was performed in 31 cases (60%), while TCZ was initiated by the hospital/ICU personnel in 21 cases (40%). Such consultation was not performed in any of the 52 control cases. Out of the 52 TCZ-treated patients, 19 (37%) received one, while 33 (63%) received two 8 mg/kg IV doses. In addition, TCZ was administered to 28 patients (54%) already in the general ward, while TCZ treatment was carried out in the ICU in 24 cases (46%) (Table 1).

In general, we did not find any differences in safety in the TCZ-treated versus non-treated group. No TCZ-related side effects were observed in patients receiving either one or two doses.

### 3.2. Effects of TCZ Treatment on COVID-19 Outcomes and MIS Biomarkers

Table 1 includes the comparison of TCZ-treated and control patients with respect to clinical outcomes. The need for ICU admission was 65% vs. 81% (*p* = 0.146), the need for ventilation was 63% vs. 77% (*p* = 0.204), the total duration of hospitalisation was 17.1 vs. 15.8 days (*p* = 0.515), and death occurred in 44% vs. 63% (*p* = 0.040), respectively (Table 1).

Figure 1 shows changes in biomarkers before and after TCZ treatment, as well as in TCZ-untreated controls. There were increased total WBC counts at baseline in the TCZ-treated group (TCZ-0) compared to controls (*p* < 0.001), which did not change significantly after TCZ therapy (TCZ-1) (*p* < 0.001 vs. controls and *p* = 0.234 vs. TCZ-0; Figure 1A). There were higher absolute neutrophil counts in the TCZ-treated group at baseline (TCZ-0; *p* < 0.001) and after treatment (TCZ-1; *p* < 0.001) in comparison to controls. There was no difference in neutrophil counts between TCZ-0 and TCZ-1 (*p* = 0.222; Figure 1B). There were low absolute lymphocyte counts in controls and in TCZ-0, which significantly increased in TCZ-1 (*p* = 0.035 vs. controls and *p* = 0.037 vs. TCZ-0; Figure 1C). Neutrophil-to-lymphocyte ratios were significantly higher in the TCZ-treated group vs. controls at baseline (TCZ-0; *p* = 0.005), which did not change upon TCZ treatment (TCZ-1; *p* = 0.455 vs. TCZ-0 and *p* < 0.001 vs. controls; Figure 1D). TCZ treatment (TCZ-1) significantly increased platelet counts compared to TCZ-0 (*p* = 0.003) and controls (*p* = 0.004) (Figure 1E). CRP was similar in the TCZ-treated and TCZ-untreated groups at baseline. TCZ treatment (TCZ-1) significantly decreased CRP levels in comparison to baseline levels in controls (*p* < 0.001) and in TCZ-0 (*p* < 0.001) (Figure 1F). Elevated levels of ferritin were observed in TCZ-0 compared to controls (*p* = 0.014). Ferritin levels in TCZ-1 were significantly lower compared to TCZ-0 (*p* = 0.043; Figure 1G). There was increased LDH in TCZ-0 compared to controls at baseline (*p* = 0.026), which did not change significantly upon TCZ treatment (TCZ-1; *p* = 0.229 vs. TCZ-0 and *p* = 0.225 vs. controls; Figure 1H). Finally, D-dimer levels were not different between TCZ-0, TCZ-1, and controls (Figure 1I).

We also compared patients receiving TCZ in the general ward (*n* = 28) to those treated with TCZ in the ICU (*n* = 24; Table 2). The two groups were similar with respect to age and symptom duration. However, the sex distribution among patients treated in the general ward was 50%-50%, while in the ICU, 21 patients (88%) were males (*p* = 0.017). Consultation with a rheumatologist occurred in 79% vs. 38% of cases (*p* = 0.005). Among patients treated in the general ward, only 36% required admission to the ICU later, and also 36% needed ventilation later (*p* < 0.001). Only 7% of those receiving TCZ in the general ward died in comparison to 88%, who died in the ICU (*p* < 0.001). At baseline, PaO_2_ (*p* = 0.046) and SaO_2_ (*p* = 0.019) were significantly higher, WBC (*p* < 0.001) and neutrophil counts (*p* = 0.002), as well as neutrophil-to-lymphocyte ratio were lower (*p* = 0.017), LDH (*p* = 0.016), and D-dimer (*p* < 0.001) were also lower among patients receiving TCZ in the general ward. After treatment, WBC (*p* = 0.027), neutrophil counts (*p* = 0.013), and neutrophil-to-lymphocyte ratio (*p* = 0.025) remained lower, platelet count became higher (*p* = 0.039), while ferritin (*p* = 0.039), LDH (*p* = 0.019), and D-dimer (*p* = 0.008) also became lower when TCZ was administered in the general ward vs. ICU (Table 2).

### 3.3. Correlations between Clinical, Laboratory, and Imaging Parameters before and after TCZ Treatment

We assessed associations between clinical outcomes, as well as laboratory and imaging markers in the TCZ-treated (Table 3 and control groups (Table 4) by Spearman’s correlation analysis. All significant correlations are shown in Table 2. In the TCZ-treated group (Table 3), in general, a consultation with a rheumatologist and two doses of TCZ vs. one dose were associated with significantly better clinical outcomes, including the need for ICU admission, ventilation, and death (*p* < 0.05) (Table 3). Increased total WBC and neutrophil counts, higher neutrophil-to-lymphocyte ratios, lower lymphocyte and platelet counts, elevated CRP, PCT, LDH, D-dimer, and IL-6 levels, known as MIS biomarkers, were all correlated with unfavourable clinical outcomes (*p* < 0.05) (Table 3). Both lower baseline PaO_2_ and SaO_2_ correlated with variable MIS biomarkers, as well as CTSS (*p* < 0.05) (Table 3). SaO_2_ was also inversely correlated with fatal outcome (*p* = 0.025). CTSS mostly correlated with elevated D-dimer and IL-6 levels, as well as lower oxygenation (*p* < 0.05). Pre- and/or post-treatment WBC, neutrophil, and platelet counts, neutrophil-to-lymphocyte ratios, as well as CRP, ferritin, PCT, LDH, D-dimer, IL-6, and CTSS variably correlated with each other (*p* < 0.05), suggesting the panel of these biomarkers might be suitable to assess MIS and to monitor TCZ treatment effects on severe COVID-19-associated MIS (Table 3). 

Administration of TCZ in the ICU vs. general ward was also correlated with other parameters (Table 3). More patients were treated in the general ward after consultation with a rheumatologist (*p* = 0.004). Pre- and/or post-treatment WBC and neutrophil counts, neutrophil-to-lymphocyte ratios, ferritin, PCT, LDH, D-dimer, and IL-6 were positively associated with TCZ treatment carried out in the ICU compared to the general ward (*p* < 0.05). In addition, lower platelet counts, PaO_2,_ and SaO_2_ were observed when treatment was performed in the ICU vs. general ward (*p* < 0.05). Finally, the need for ventilation and mortality were also higher in ICU-admitted patients (*p* < 0.001; Table 3). 

As discussed above, patients in the control group were in a less inflammatory state in COVID-19. Here, thrombocytopenia was associated with unfavourable outcomes (ICU, ventilation, death; *p* < 0.05), while CTSS correlated with the need for ventilation (*p* = 0.021) (Table 4). MIS biomarkers including neutrophilia, lymphocytopenia, high neutrophil-to-lymphocyte ratios, thrombocytopenia, elevated CRP, PCT, LDH, D-dimer, and ferritin variably and significantly correlated with each other, as well as with impaired PaO_2_ and SaO_2_. Thus, these biomarkers are also suitable to monitor systemic inflammation and associated pathologies in COVID-19 patients with less extensive MIS (Table 4). 

## 4. Discussion

In this single-centre study of 104 patients with COVID-19 pneumonia admitted to hospital, we compared corticosteroid + TCZ-treated (*n* = 52) and only corticosteroid treated (control; *n* = 52) patient subsets with respect to laboratory biomarkers of MIS and CTSS. 

At the time of admission, patients selected for TCZ therapy (TCZ-0) had more pronounced MIS indicated by higher WBC and neutrophil counts, neutrophil-to-lymphocyte rations, as well as increased IL-6, ferritin, and LDH levels compared to controls. Increased WBC and neutrophil counts, lymphocytopenia, increased neutrophil-to-lymphocyte ratio, thrombocytopenia, elevated CRP, ferritin, D-dimer, and IL-6 levels have been associated with MIS and unfavourable clinical outcomes in COVID-19 [5,6,8]. We and others have also evaluated CTSS as a useful scoring system in determining more severe COVID-19 pneumonia and worse outcomes [25,26]. 

Regarding clinical outcomes, TCZ treatment was associated with significantly lower mortality compared to controls. In the TCZ-treated group, there was also a tendency of less ICU admission and less need for ventilation. Indeed, when administered to severe COVID-19 patients with high-degree MIS in combination with corticosteroids, TCZ improved survival [11,12,13]. However, TCZ compared to placebo did not improve survival in other studies [19]. More than 50% of patients received TCZ treatment in the general ward, while the rest were treated in the ICU. When comparing these two patient subsets, significantly better outcomes including later admission to ICU (36% vs. 100%), need for ventilation (36% vs. 100%), and death (7% vs. 88%) could be reached when TCZ was administered already in the general ward compared to those receiving TCZ in the ICU. This also required close consultation with a rheumatologist. 

In our study, TCZ treatment (TCZ-1) significantly improved absolute lymphocyte and platelet counts, and decreased CRP and ferritin levels in comparison to baseline (TCZ-0). Thus, TCZ improved most biomarkers of COVID-19-associated MIS. Total WBC, absolute neutrophil counts, neutrophil-to-lymphocyte ratios, LDH, and D-dimer remained unchanged after TCZ treatment (TCZ-1) compared to pre-treatment levels (TCZ-0). Increased neutrophil-to-lymphocyte ratio is also a useful biomarker of MIS [5,6,8]. LDH and D-dimer are markers of lung damage and microthrombosis, respectively [5,27], which might not be influenced by TCZ, a compound that primarily attenuates IL-6 effects in inflammation. 

Finally, we correlated clinical outcomes, laboratory, and imaging biomarkers in the TCZ-treated group, as well as in controls. We present data suggesting that consultation with a rheumatologist by the hospital/ICU personnel might improve clinical outcomes. We also confirmed that two doses of TCZ yield better clinical outcomes compared to one dose. In contrast to two reports suggesting more safety issues with multiple TCZ doses compared to one dose [28,29], we did not observe any safety concerns with one or two doses of TCZ. We, similarly to others [5,6,8,27], confirmed that leucocytosis, neutrophilia, lymphocytopenia, thrombocytopenia, increased CRP, PCT, ferritin, LDH, D-dimer, and IL-6 levels, as well as impaired oxygenation are suitable indicators for MIS, and might be associated with worse clinical outcomes. CTSS is a useful scoring system for the determination of lung involvement and damage in COVID-19 pneumonia [25,26]. Thus, a panel of laboratory biomarkers described above together with PaO_2_, SaO_2_ and, if available, CTSS might be useful to determine the severity of COVID-19 pneumonia. These biomarkers might also be suitable to monitor the effects of anti-inflammatory therapies, such as TCZ. As described above, patients eventually undergoing TCZ therapy had more severe COVID-19 than controls. All these results support that TCZ should be used in the highly inflammatory stages of severe COVID-19 [6,11,16,17,18,20,22,23,24]. 

This study has certain strengths and limitations. The major strength of this study is that this is the first Hungarian study assessing the value of TCZ therapy in conjunction with several clinical, laboratory, and imaging biomarkers of COVID-19 pneumonia. We also pointed out that the managing physician should collaborate with a rheumatologist having expertise in TCZ treatment. Possible limitations may include the single-centre nature of the study and the relatively low number of patients. In addition, we only determined CTSS at baseline, so we did not have follow-up CTSS data. We have not validated our findings against other cohorts, and we have not considered population-specific biases.

Of course, the place of TCZ in the treatment of COVID-19 is not yet fully clear. There has still been some controversy with respect to monotherapy versus combination with corticosteroids; the timing of TCZ treatment during the course of COVID-19; the attributes of patients likely to respond to TCZ; as well as cost-effectiveness of TCZ therapy [14,21]. 

## 5. Conclusions

In conclusion, TCZ was successfully introduced to the therapy for severe COVID-19 pneumonia in Hungary. TCZ treatment was found to be safe in our COVID-19 pneumonia patients. TCZ in combination with corticosteroids favourably influenced various laboratory biomarkers of MIS, and TCZ + corticosteroid treatment was associated with better outcomes compared to controls. Patients receiving TCZ treatment already in the general ward exerted much better outcomes than those treated in the ICU. We also pointed out the importance of the involvement of a rheumatologist in making decisions about the initiation of TCZ treatment in severe COVID-19 pneumonia.

## Figures and Tables

**Figure 1 biomedicines-11-00349-f001:**
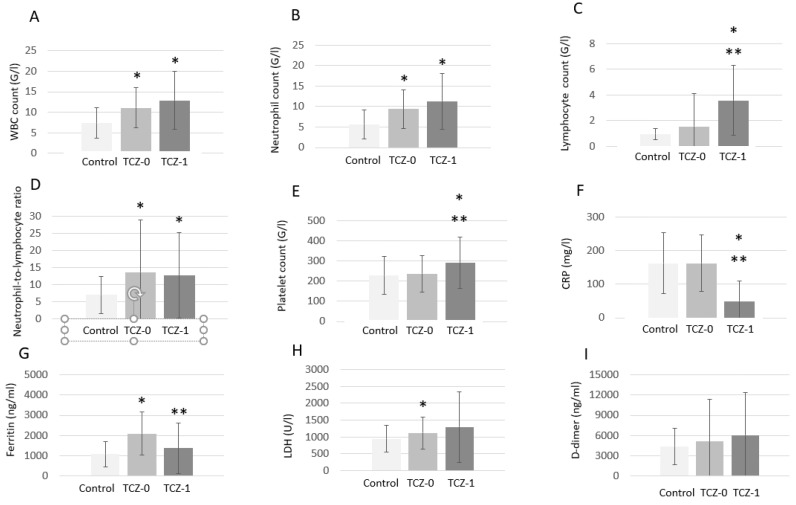
Laboratory biomarkers in TCZ-treated (pre- [TCZ-0] and post-treatment [TCZ-1]) and control patients. (**A**) white blood cell (WBC) count; (**B**) neutrophil count; (**C**) lymphocyte count; (**D**) neutrophil-to-lymphocyte ratio; (**E**) platelet count; (**F**) C-reactive protein (CRP) level; (**G**) ferritin level; (**H**) lactate dehydrogenase (LDH) level; (**I**) D-dimer level. *, *p* < 0.05 vs. control patients; **, *p* < 0.05 vs. TCZ-0.

**Table 1 biomedicines-11-00349-t001:** Baseline characteristics of TCZ-treated and control patients.

Parameters	TCZ-Treated	Control (TCZ-Untreated)	*p* Value *
**General characteristics**			
Number, *n*	52	52	1.000
Age, mean years ± SD (range) ^#^	60.2 ± 9.6 (37–78)	60.1 ± 9.8 (40–76)	0.968
Female:male ratio, *n*	17:35	17:35	0.855
Symptom duration until hospital admission, mean days ± SD (range) ^#^	9.75 ± 3.66 (1–17)	9.65 ± 3.46 (1–15)	0.877
Consultation with a rheumatologist, *n* (%) ^#^	31 (60)	0 (0)	** *<0.001* **
**Treatments**			
TCZ doses, *n* (%) 1 dose 2 doses	19 (37) 33 (63)	- -	- -
TCZ administration, *n* (%) General ward ICU	28 (54) 24 (46)	- -	- -
Antiviral therapy, *n* (%) ^#^	52 (100)	52 (100)	1.000
Corticosteroid treatment, *n* (%) ^#^	52 (100)	52 (100)	1.000
**Outcome measures**			
ICU admission, *n* (%)	34 (65)	42 (81)	0.146
Ventilation, *n* (%)	33 (63)	40 (77)	0.204
Death, *n* (%)	23 (44)	33 (63)	** *0.040* **
Total duration of hospitalisation, mean days ± SD	17.1 ± 7.3	15.8 ± 8.1	0.515
Duration of hospitalisation after TCZ initiation, mean days ± SD	14.5 ± 7.8	-	-
**Laboratory biomarkers**			
PaO_2_ (mmHg), mean ± SD ^#^	50. 4 ± 6.9	54.7 ± 10.9	0.071
SaO_2_ (%), mean ± SD ^#^	83.0 ± 8.5	86.6 ± 8.5	0.080
Total WBC count (G/l), mean ± SD ^#^	11.10 ± 4.86	7.38 ± 3.71	** *<0.001* **
Absolute neutrophil count (G/l), mean ± SD ^#^	9.37 ± 4.68	5.67 ± 3.51	** *<0.001* **
Absolute lymphocyte count (G/l), mean ± SD ^#^	1.53 ± 2.59	0.96 ± 0.44	0.118
Neutrophil-to-lymphocyte ratio, mean ± SD ^#^	13.6 ± 15.4	7.0 ± 5.4	** *0.005* **
Absolute platelet count (G/l), mean ± SD ^#^	236.6 ± 90.7	228.2 ± 93.4	0.668
CRP (mg/l), mean ± SD ^#^	162.0 ± 83.7	162.5 ± 91.1	0.725
IL-6 (pg/mL), mean ± SD ^#^	327.5 ± 473.5	91.7 ± 113.2	** *0.003* **
Ferritin (ng/mL), mean ± SD ^#^	2100.3 ± 3065.6	1075.3 ± 637.1	** *0.026* **
PCT (ng/mL), mean ± SD ^#^	0.08 ± 0.34	0.22 ± 0.17	0.081
LDH (U/l), mean ± SD ^#^	1116.5 ± 471.2	941.4 ± 394.2	** *0.020* **
D-dimer (ng/mL), mean ± SD ^#^	5115.7 ± 6238.0	4384.1 ± 2718.0	0.275
**Imaging**			
CTSS score, mean *n* ± SD ^#^	19.9 ± 4.0	20.2 ± 4.2	0.899

^#^ Baseline parameters. * Significant differences are in bold italics. Abbreviations: CRP, C-reactive protein; CT, computed tomography; CTSS, CT severity score; ICU, intensive care unit; IL, interleukin; LDH, lactate dehydrogenase; PaO_2_, partial oxygen pressure; PCT, procalcitonin; SaO_2_, oxygen saturation; SD, standard deviation; TCZ, tocilizumab; WBC, white blood cell.

**Table 2 biomedicines-11-00349-t002:** Comparison of TCZ-treated patients receiving TCZ in the general ward versus ICU.

Parameters	TCZ Received in General Ward	TCZ Received in ICU	*p* Value *
**General characteristics**			
Number, *n*	28	24	-
Age, mean years ± SD ^#^	61.9 ± 8.2	58.4 ± 10.9	0.238
Female:male ratio, *n*	14:14	3:21	** *0.017* **
Symptom duration until hospital admission, mean days ± SD ^#^	9.63 ± 3.93	9.58 ± 3.57	0.971
Consultation with a rheumatologist, *n* (%) ^#^	22 (79)	9 (38)	** *0.005* **
**Outcome measures**			
ICU admission, *n* (%)	10 (36)	24 (100)	** *<0.001* **
Ventilation, *n* (%)	10 (36)	24 (100)	** *<0.001* **
Death, *n* (%)	2 (7)	21 (88)	** *<0.001* **
Total duration of hospitalisation, mean days ± SD	16.7 ± 6.7	15.8 ± 7.5	0.645
**Laboratory biomarkers**			
PaO_2_-0 (mmHg), mean ± SD	52.1 ± 6.6	48.6 ± 7.4	** *0.046* **
SaO_2_-0 (%), mean ± SD	86.1 ± 5.8	79.8 ± 9.8	** *0.019* **
WBC-0 (G/l), mean ± SD	8.71 ± 2.94	13.6 ± 5.1	** *<0.001* **
WBC-1 (G/l), mean ± SD	10.6 ± 3.8	15.5 ± 8.8	** *0.027* **
NEU-0 (G/l), mean ± SD	7.17 ± 2.58	11.5 ± 5.1	** *0.002* **
NEU-1 (G/l), mean ± SD	8.68 ± 3.03	14.0 ± 8.4	** *0.013* **
LYMP-0 (G/l), mean ± SD	1.46 ± 1.18	0.96 ± 0.69	0.097
LYMP-1 (G/l), mean ± SD	1.64 ± 2.01	1.14 ± 0.71	0.325
NEU:LYMP-0, mean ± SD	7.75 ± 7.35	18.0 ± 17.6	** *0.017* **
NEU:LYMP-1, mean ± SD	9.63 ± 7.16	16.8 ± 14.1	** *0.025* **
PLT-0 (G/l), mean ± SD	224.2 ± 85.9	245.7 ± 98.5	0.473
PLT-1 (G/l), mean ± SD	318.6 ± 87.8	235.6 ± 137.5	** *0.039* **
CRP-0 (mg/l), mean ± SD	143.0 ± 74.1	184.8 ± 86.4	0.078
CRP-1 (mg/l), mean ± SD	43.6 ± 67.7	51.9 ± 49.3	0.658
IL6-0 (pg/mL), mean ± SD	303.2 ± 659.3	375.9 ± 288.2	0.659
FERR-0 (ng/mL), mean ± SD	1318.8 ± 1040.8	2940.4 ± 3670.2	0.074
FERR-1 (ng/mL), mean ± SD	871.4 ± 613.2	1950.8 ± 1624.0	** *0.039* **
PCT-0 (ng/mL), mean ± SD	<0.001	0.08 ± 0-28	0.162
PCT-1 (ng/mL), mean ± SD	<0.001	1.35 ± 4.00	0.148
LDH-0 (U/l), mean ± SD	1023.3 ± 446.4	1366.4 ± 474.9	** *0.016* **
LDH-1 (U/l), mean ± SD	773.9 ± 379.7	2032.5 ± 1542.8	** *0.019* **
DD-0 (ng/mL), mean ± SD	1021.1 ± 1048.4	8137.2 ± 6731.6	** *<0.001* **
DD-1 (ng/mL), mean ± SD	3155.2 ± 5040.1	8987.2 ± 6731.6	** *0.008* **
**Imaging**			
CTSS-0, mean *n* ± SD ^#^	19.2 ± 4.3	21.1 ± 2.9	0.176

^#^ Baseline parameters. * Significant differences are in bold italics. Abbreviations and explanations: 0, pre-treatment; 1, post-treatment; CRP, C-reactive protein; CT, computed tomography; CTSS, CT severity score; DD, D-dimer; FERR, ferritin; ICU, intensive care unit; IL6, interleukin 6; LDH, lactate dehydrogenase; LYMP, lymphocyte, NEU, neutrophil; NEU:LYMP, neutrophil-to-lymphocyte ratio; PaO_2_, partial oxygen pressure; PCT, procalcitonin; PLT, platelets; SaO_2_, oxygen saturation; SD, standard deviation; TCZ, tocilizumab; WBC, white blood cell.

**Table 3 biomedicines-11-00349-t003:** Significant correlations between various parameters in the TCZ-treated and the control groups. * TCZ-treated group.

	CONS	DOSE	WBC 0	WBC 1	NEU 0	NEU 1	LYM 0	NEU/LY 0	NEU/LY 1	PLT 1	CRP 0	CRP 1	FERR 0	FERR 1	PCT 0	PCT 1	LDH 0	LDH 1	DD 0	DD 1	IL6 0	PaO_2_ 0	SaO_2_ 0	CTSS 0	ICU	**VENT**	**DEATH**
**ADMIN IN ICU**	R = −0.388 *p* = 0.004		R = 0.479 *p* < 0.001	R = 0.282 *p* = 0.045	R = 0.436 *p* = 0.001	R = 0.297 *p* = 0.035		R = 0.411 *p* = 0.002		R = −0.406 *p* = 0.003			R = 0.421 *p* = 0.003	R = 0.572 *p* < 0.001		R = 0.370 *p* = 0.009	R = 0.399 *p* = 0.004	R = 0.484 *p* = 0.002	R = 0.651 *p* < 0.001	R = 0.471 pí0.001	R = 0.337 *p* = 0.015	R = −0.292 *p* = 0.041	R = −0.336 *p* = 0.014		R = 0.652 *p* < 0.001	R = 0.601 *p* < 0.001	R = 0.772 *p* < 0.001
**FERR-0**											R = 0.331 *p* = 0.023																
**FERR-1**				R = 0.590 *p* < 0.001		R = 0.579 *p* < 0.001																					
**PCT-0**		R = −0.296 *p* = 0.032	R = 0.314 *p* = 0.022		R = 0.362 *p* = 0.008			R = 0.390 *p* = 0.004	R = 0.376 *p* = 0.006			R = 0.471 *p* < 0.001															
**PCT-1**			R = 0.414 *p* = 0.003	R = 0.347 *p* = 0.015	R = 0.444 *p* = 0.001	R = 0.356 *p* = 0.012						R = 0.347 *p* = 0.015		R = 0.730 *p* < 0.001													
**LDH-0**			R = 0.296 *p* = 0.037									R = 0.284 *p* = 0.048															
**LDH-1**			R = 0.469 *p* = 0.003	R = 0.511 *p* < 0.001	R = 0.518 *p* < 0.001	R = 0.560 *p* < 0.001		R = 0.563 *p* < 0.001	R = 0.376 *p* = 0.018	R = −0.515 *p* < 0.001	R = 0.323 *p* = 0.045	R = 0.423 *p* = 0.007		R = 0.794 *p* < 0.001	R = 0.395 *p* = 0.013	R = 0.862 *p* < 0.001											
**DD-0**			R = 0.765 *p* < 0.001	R = 0.528 *p* < 0.001	R = 0.772 *p* < 0.001	R = 0.554 *p* < 0.001		R = 0.433 *p* = 0.003	R = 0.380 *p* = 0.009	R = −0.484 *p* = 0.001			R = 0.328 *p* = 0.034	R = 0.358 0.048	R = 0.314 *p* = 0.038	R = 0.440 *p* = 0.004		R = 0.614 *p* < 0.01									
**DD-1**			R = 0.470 *p* = 0.003	R = 0.528 *p* < 0.001	R = 0.370 *p* = 0.024	R = 0.359 *p* = 0.029			R = 0.421 *p* = 0.004	R = −0.606 *p* < 0.001	R = 0.414 *p* = 0.011	R = 0.517 *p* = 0.001		R = 0.358 *p* = 0.048	R = 0.358 *p* = 0.030	R = 0.376 *p* = 0.024	R = 0.366 *p* = 0.031	R = 0.579 *p* < 0.001									
**IL6-0**			R = 0.562 *p* < 0.001	R = 0.355 *p* = 0.012	R = 0.376 *p* = 0-007	R = 0.379 *p* = 0.007						R = 0.375 *p* = 0.008			R = 0.376 *p* = 0.007	R = 0.515 *p* < 0.001	R = 0.316 *p* = 0.029	R = 0.623 *p* < 0.01	R = 0.462 *p* = 0.002	R = 0.596 *p* < 0.001							
**PaO_2_-0**			R = −0.311 *p* = 0.025					R = −0.369 *p* = 0.007												R = −0.309 *p* = 0.046							
**SaO_2_-0**	R = 0.362 *p* = 0.008		R = −0.390 *p* = 0.004		R = −0.340 *p* = 0.013			R = −0.558 *p* < 0.001	R = −0.309 *p* = 0.013	R = −0.294 *p* = 0.033	R = −0.294 *p* = 0.033								R = −0.440 *p* = 0.003	R = −0.332 *p* = 0.045		R = 0.873 *p* < 0.001					
**CTSS-0**																				R = 0.398 *p* = 0.022	R = 0.294 *p* = 0.038	R = −0.448 *p* = 0.002	R = −0.416 *p* = 0.004				
**ICU**	R = −0.281 *p* = 0.042		R = 0.367 *p* = 0.007		R = 0.368 *p* = 0.007			R = 0.329 *p* = 0.016	R = 0.270 *p* = 0.049	R = −0.305 *p* = 0.029							R = 0.282 *p* = 0.047	R = 0.404 *p* = 0.011	R = 0.372 *p* = 0.013	R = 0.324 *p* = 0.049	R = 0.369 *p* = 0.008						
**VENT**	R = −0.310 *p* = 0.024	R = −0.287 *p* = 0.043	R = 0.404 *p* = 0.003		R = 0.410 *p* = 0.002			R = 0.355 *p* = 0.009	R = 0.297 *p* = 0.031	R = −0.313 *p* = 0.025								R = 0.405 *p* = 0.011	R = 0.396 *p* = 0.008	R = 0.372 *p* = 0.023	R = 0.395 *p* = 0.004				R = 0.959 *p* < 0.001		
**DEATH**	R = −0.388 *p* = 0.004	R = −0.384 *p* = 0.004	R = 0.480 *p* < 0.001	R = 0.390 *p* = 0.005	R = 0.406 *p* = 0.003	R = 0.406 *p* = 0.003	R = −0.255 *p* = 0.048	R = 0.433 *p* = 0.001	R = 0.439 *p* = 0.001	R = −0.271 *p* = 0.048	R = 0.272 *p* = 0.047				R = 0.283 *p* = 0.048	R = 0.297 *p* = 0.043		R = 0.479 *p* = 0.002	R = 0.588 *p* < 0.001	R = 0.515 *p* = 0.001	R = 0.436 *p* = 0.002		R = −0.308 *p* = 0.025	R = 0.286 *p* = 0.038	R = 0.652 *p* < 0.001	R = 0.680 *p* < 0.001	

* Significant correlations only.

**Table 4 biomedicines-11-00349-t004:** Significant correlations between various parameters in the control groups.

	NEU 0	LYM 0	NEU/LY 0	PLT 0	CRP 0	FERR 0	LDH 0	DD 0	PaO_2_ 0	CTSS 0	ICU	VENT
**FERR-0**												
**PCT-0**	R = 0.309 *p* = 0.035		R = 0.377 *p* = 0.010									
**LDH-0**						R = 0.566 *p* < 0.001						
**DD-0**		R = −0.355 *p* = 0.021				R = 0.358 *p* = 0.030						
**IL6-0**												
**PaO_2_-0**					R = −0.312 *p* = 0.047			R = −0.391 *p* = 0.024				
**SaO_2_-0**							R = −0.347 *p* = 0.023		R = 0.616 *p* < 0.001			
**CTSS-0**							R = 0.324 *p* = 0.022					
**ICU**				R = −0.405 *p* = 0.003								
**VENT**				R = −0.428 *p* = 0.002						R = 0.318 *p* = 0.021	R = 0.891 *p* < 0.001	
**DEATH**				R = −0.400 *p* = 0.003							R = 0.542 *p* < 0.001	R = 0.627 *p* < 0.001

Abbreviations and explanations: 0, pre-treatment; 1, post-treatment; ADMIN, administration of TCZ; CONS, consultation with rheumatologist; CRP, C-reactive protein; CT, computed tomography; CTSS, CT severity score; DD, D-dimer; FER, ferritin; ICU, intensive care unit; IL, interleukin; LDH, lactate dehydrogenase; LYM, lymphocyte count; NEU, neutrophil count; NEU/LY, neutrophil-to-lymphocyte ratio; PaO_2_, partial oxygen pressure; PCT, procalcitonin; PLT, platelet count; SaO_2_, oxygen saturation; TCZ, tocilizumab; VENT, ventilation; WBC, white blood cell count.

## Data Availability

The provided data can be found in the Borsod Academic County Hospital’s electronic medical system.

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
