# Peer review of "Tocilizumab in Combination with Corticosteroids in COVID-19 Pneumonia: A Single-Centre Retrospective Controlled Study"

_biomedicines, 2023, doi:10.3390/biomedicines11020349_

Round 1
Reviewer 1 Report
The manuscript on “Tocilizumab in COVID-19 pneumonia: a single-center retrospective-controlled study” is interesting and used as a tool for the treatment of pneumonia in Covid patients.
As per the conclusions of the study, the treatment considered as combination therapy of TCZ with corticosteroids. Rewrite the title. Further, similar study reported by Rosas et al., 2021, comparison with placebo.
Name the corticosteroids (trade name, company, dose, etc) used for the study.
Line#181, The full patient cohort included 70 men and 34 women, 35 men and 17 women in both groups.
Not understanding statement.
Figure 1 – improve the resolution for clarity.
What is the basis for selection of TCZ dose for the study. Write the reference.
Is there any specific TCZ dose related side effects observed during the study. How the side effects level of severity in ICU vs normal general ward patients.
Write the dosage regimen details – time of administration, route, length, etc.
Author Response
We thank the reviewer for carefully reviewing our manuscript and for the very helpful comments.
The manuscript on “Tocilizumab in COVID-19 pneumonia: a single-center retrospective-controlled study” is interesting and used as a tool for the treatment of pneumonia in Covid patients.
Many thanks for the positive opinion.
As per the conclusions of the study, the treatment considered as combination therapy of TCZ with corticosteroids. Rewrite the title. Further, similar study reported by Rosas et al., 2021, comparison with placebo.
Many thanks. In Hungary the combination of TCZ with corticosteroids is also standard of care. We added corticosteroids to the title, abstract and throughout the text. We added the paper of Rosas et al.
Name the corticosteroids (trade name, company, dose, etc) used for the study.
We added the names and brand names to the text and also the dosing regimen.
Line#181, The full patient cohort included 70 men and 34 women, 35 men and 17 women in both groups.
Sorry, this was a mistake, we now corrected the numbers.
Figure 1 – improve the resolution for clarity.
Figure improved.
What is the basis for selection of TCZ dose for the study. Write the reference.
The 8 mg/kg dose was used in most studies, we added the references. We now also added the dosing regimen.
Is there any specific TCZ dose related side effects observed during the study. How the side effects level of severity in ICU vs normal general ward patients.
As now added to the text, we did not notice any side effects due to TCZ.
Write the dosage regimen details – time of administration, route, length, etc.
We now added the dosing regimen
Reviewer 2 Report
The research entitled "Tocilizumab in COVID-19 pneumonia: a single-centre retrospective controlled study" is a well performed research with significantly important results.
Before the publication, I would like to suggest some points which can be addressed.
First of all, the title: as the administration of TCZ was done in the patients treated with corticosteroids ,
So it will be more appropriate to highlight this in the title of the article.
Similarly, in the discussion and results it will be mor appropriate to describe the clinical outcomes while conjugating with the corticosteroids. As the clinical outcomes are the additive effects of both drugs.
Line no 138: check the figure?
In the discussion authors must provide the limitations of the study.
Additionally, the limitation of the use of TCZ must be provided in the discussion at the end.
The Contradictions on the efficacy and safety of TCZ section of the following article can be utilised
(http://dx.doi.org/10.7324/JAPS.2021.1101104)
- What is the best time to start using MABs such as TCZ that target both soluble and membrane-bound receptors?
- What are the attributes of patients which can be treated with TCZ? Are these only severely infected patients with COVID-19?
- What is the efficacy and risk status of TCZ in patients at risk of other deadly infections, including bacterial and fungal, and viral infections?
- Will TCZ be used to treat COVID-19 patients at a reasonable cost?
- [http://dx.doi.org/10.7324/JAPS.2021.1101104].
The conclusion was provided as the following with the discussion:
In conclusion, TCZ was successfully introduced to the therapy of severe COVID-19 pneumonia in Hungary. TCZ favourably influenced various laboratory biomarkers of MIS and TCZ treatment was associated with better outcomes compared to controls. Patients receiving TCZ treatment already in the general ward exerted much better outcomes than those treated in the ICU. We also pointed out the importance of the involvement of rheumatologist in making decision about the initiation of TCZ treatment in severe COVID-19 pneumonia.
I personally, believe that the conclusion must be provided separately, while providing certain important outcomes of your study in the form of data and then describe it briefly,
Author Response
We thank the reviewer for carefully reviewing our manuscript and for the very helpful comments.
The research entitled "Tocilizumab in COVID-19 pneumonia: a single-centre retrospective controlled study" is a well performed research with significantly important results.
Many thanks for this very positive opinion.
Before the publication, I would like to suggest some points which can be addressed.
First of all, the title: as the administration of TCZ was done in the patients treated with corticosteroids, so it will be more appropriate to highlight this in the title of the article.
Similarly, in the discussion and results it will be more appropriate to describe the clinical outcomes while conjugating with the corticosteroids. As the clinical outcomes are the additive effects of both drugs.
Many thanks. In Hungary the combination of TCZ with corticosteroids is also standard of care. We added corticosteroids to the title, abstract and throughout the text, also in the Discussion.
Line no 138: check the figure?
Sorry, this was a mistake, we now corrected the numbers.
In the discussion authors must provide the limitations of the study.
We already included strengths and limitations of the study.
Additionally, the limitation of the use of TCZ must be provided in the discussion at the end.
The Contradictions on the efficacy and safety of TCZ section of the following article can be utilised
(http://dx.doi.org/10.7324/JAPS.2021.1101104)
- What is the best time to start using MABs such as TCZ that target both soluble and membrane-bound receptors?
- What are the attributes of patients which can be treated with TCZ? Are these only severely infected patients with COVID-19?
- What is the efficacy and risk status of TCZ in patients at risk of other deadly infections, including bacterial and fungal, and viral infections?
- Will TCZ be used to treat COVID-19 patients at a reasonable cost?
- [http://dx.doi.org/10.7324/JAPS.2021.1101104].
The conclusion was provided as the following with the discussion:
In conclusion, TCZ was successfully introduced to the therapy of severe COVID-19 pneumonia in Hungary. TCZ favourably influenced various laboratory biomarkers of MIS and TCZ treatment was associated with better outcomes compared to controls. Patients receiving TCZ treatment already in the general ward exerted much better outcomes than those treated in the ICU. We also pointed out the importance of the involvement of rheumatologist in making decision about the initiation of TCZ treatment in severe COVID-19 pneumonia.
I personally, believe that the conclusion must be provided separately, while providing certain important outcomes of your study in the form of data and then describe it briefly,
Many thanks, we now included the reference mentioned with the major statements. We also have Conclusions separately now and expanded this section as requested.
Round 2
Reviewer 1 Report
Manuscript modified as per the suggested edits. The comments addressed were also satisfactory.
Reviewer 2 Report
The authors have sufficiently revised the manuscript. It can be accepted for the publication.
Best Wishes.